# Desaturase Activity and the Risk of Type 2 Diabetes and Coronary Artery Disease: A Mendelian Randomization Study

**DOI:** 10.3390/nu12082261

**Published:** 2020-07-28

**Authors:** Susanne Jäger, Rafael Cuadrat, Per Hoffmann, Clemens Wittenbecher, Matthias B. Schulze

**Affiliations:** 1Department of Molecular Epidemiology, German Institute of Human Nutrition Potsdam-Rehbruecke, 14558 Nuthetal, Germany; susanne.jaeger@dife.de (S.J.); Rafael.Cuadrat@dife.de (R.C.); cwittenbecher@hsph.harvard.edu (C.W.); 2German Center for Diabetes Research (DZD), 85764 Neuherberg, Germany; 3Human Genomics Research Group, Department of Biomedicine, University of Basel, 4031 Basel, Switzerland; per.hoffmann@unibas.ch; 4Institute of Human Genetics, University of Bonn, School of Medicine & University Hospital Bonn, 53105 Bonn, Germany; 5Department of Nutrition, Harvard T.H. Chan School of Public Health, Boston, MA 02115, USA; 6Institute of Nutritional Science, University of Potsdam, 14558 Nuthetal, Germany

**Keywords:** Δ5-desaturase, Δ6-desaturase, type 2 diabetes, coronary artery disease, Mendelian randomization, multivariable Mendelian randomization, *FADS*-gene-cluster, fatty acids

## Abstract

Estimated Δ5-desaturase (D5D) and Δ6-desaturase (D6D) are key enzymes in metabolism of polyunsaturated fatty acids (PUFA) and have been associated with cardiometabolic risk; however, causality needs to be clarified. We applied two-sample Mendelian randomization (MR) approach using a representative sub-cohort of the European Prospective Investigation into Cancer and Nutrition (EPIC)–Potsdam Study and public data from DIAbetes Genetics Replication And Meta-analysis (DIAGRAM) and Coronary ARtery DIsease Genome wide Replication and Meta-analysis (CARDIoGRAM) genome-wide association studies (GWAS). Furthermore, we addressed confounding by linkage disequilibrium (LD) as all instruments from *FADS1* (encoding D5D) are in LD with *FADS2* (encoding D6D) variants. Our univariable MRs revealed risk-increasing total effects of both, D6D and D5D on type 2 diabetes (T2DM) risk; and risk-increasing total effect of D6D on risk of coronary artery disease (CAD). The multivariable MR approach could not unambiguously allocate a direct causal effect to either of the individual desaturases. Our results suggest that D6D is causally linked to cardiometabolic risk, which is likely due to downstream production of fatty acids and products resulting from high D6D activity. For D5D, we found indication for causal effects on T2DM and CAD, which could, however, still be confounded by LD.

## 1. Introduction

Δ5-Desaturase (D5D) and Δ6-desaturase (D6D) are key enzymes for the synthesis of longer chain poly-unsaturated fatty acids (PUFA) from plant-derived precursor fatty acids linoleic acid (LA) and α-linolenic acid (ALA) (see Figure 1a).

Prospective cohort studies used product-to-precursor ratios of fatty acids measured in blood fractions, representing surrogate markers of estimated liver desaturase activity [1]. Overall, study findings support that a higher estimated D6D activity (ratio of 18:3n-6/18:2n-6) is related to an increased type 2 diabetes (T2DM) risk, while the contrary is the case for estimated D5D activity (20:4n-6/20:3n-6) [1,2,3]. Furthermore, higher D6D activity is linked to higher cardiovascular mortality [4] and higher estimated D5D activity to lower cardiovascular mortality and coronary heart disease (CHD) risk [4,5].

To investigate causal relationships between risk factors and disease outcomes, genetic variants can be used as instrumental variables in Mendelian randomization (MR). The principle is based on Mendels’s second law of inheritance, stating that the assignment of alleles is random during meiosis [6].

In humans, D5D and D6D are encoded by *FADS1* and *FADS2* genes [7] and SNPs in that gene region have been reliably linked to desaturase activities [1,8,9] and cardiometabolic traits [10,11]. Recent MR studies have linked single fatty acids to T2DM risk [12] and CVD risk [13]. However, they are limited by the high correlation between the single fatty acids and hence individual effects could largely not be elucidated. A previous MR study found a direct relation for D6D activity and tended to support an inverse relation for D5D activity using a *FADS1*-SNP as genetic instrument [14]. However, the *FADS*-gene cluster is characterized by high linkage disequilibrium (LD) where all instruments from *FADS1* are in LD with *FADS2* variants [1,7]. Therefore, it is hard to select suitable genetic instruments for MR to disentangle the causal nature of each desaturase on the respective outcome without the limitation of confounding by LD. Furthermore, genetic instruments might associate with multiple exposures due to pleiotropic or mediating effects or both [15]. By applying multivariable MR, we are able to use multiple genetic instruments to estimate direct effects of both desaturases simultaneously in one model [15], providing novel insights into causal inference of desaturase activity and cardiometabolic outcomes (see Figure 1b).

In contrast to previous MR studies on PUFA metabolism, we aim to estimate the causal effects of D6D and D5D activity, instead of single fatty acids, on the risk of T2DM and CAD. To this end, we selected suitable genetic instruments in genome-wide association analyses on estimated D6D and D5D activities. Disease associations were drawn from large public genome-wide association studies (GWAS) on T2DM and coronary artery disease (CAD). In univariable and multivariable two-sample MR studies we estimated the causal effects of D6D and D5D activity on T2DM risk and CAD risk, accounting for interrelations between single desaturases and hence single fatty acids. Finally, we address the phenomenon of confounding by LD within the *FADS* gene cluster, which was not or not sufficiently accounted for in previous MR studies.

## 2. Materials and Methods

### 2.1. Study Population

#### 2.1.1. Individual-Level Data from EPIC-Potsdam

The European Prospective Investigation into Cancer and Nutrition (EPIC)-Potsdam study consists of 27,548 participants recruited between 1994 and 1998 from the general population in Potsdam and surroundings [16]. We used a random sample within the EPIC-Potsdam study, described in detail previously [17]. Briefly, a sub-cohort of 2500 individuals was randomly selected from 26,444 participants who provided blood samples at baseline. Participants with prevalent diabetes, diabetes medication at baseline or prevalent myocardial infarction were excluded. Further exclusion criteria were missing genetic data and missing or implausible data on fatty acid measurements, leaving 1853 individuals for analyses in the sub-cohort (Appendix A).

All participants provided written informed consent. The study was approved by the ethics committee of the State of Brandenburg, Germany. All procedures were in accordance with the ethical standards of the institutional and/or national research committee and with the 1964 Helsinki declaration and its later amendments or comparable ethical standards.

#### 2.1.2. Summary-Level Data from DIAGRAM and CARDIoGRAM

For outcome associations, we used summary-level data obtained from the DIAbetes Genetics Replication And Meta-analysis (DIAGRAM) consortium without BMI adjustment [18] and from meta-analysis of UK Biobank SOFT CAD GWAS with Coronary ARtery DIsease Genome wide Replication and Meta-analysis (CARDIoGRAM) plus The Coronary Artery Disease (C4D) Genetics (CARDIoGRAMplusC4D) 1000 Genomes-based GWAS and the Myocardial Infarction Genetics and CARDIoGRAM Exome [19]. The SOFT CAD phenotype incorporates self-reported angina or other evidence of chronic coronary heart disease [19].

### 2.2. DNA-Extraction, Genotyping and Quality Control

The DNA was extracted from buffy coats using the chemagic DNA Buffy Coat Kit on a Chemagic Magnetic Separation Module I (PerkinElmer chemagen Technologies, Baesweiler, Germany) according to the manufacturer’s manual. Samples from EPIC-Potsdam participants were genotyped with three different genotyping arrays: Human660W-Quad_v1_A (*n* = 355), HumanCoreExome-12v1-0_B (*n* = 622) and Illumina Infinium OmniExpressExome-8v1-3_A DNA Analysis BeadChip (*n* = 1349). Genotyping and quality control of the Human660W-Quad_v1_A and HumanCoreExome-12v1-0_B chips was described elsewhere [20]. Genotyping using the Illumina Infinium OmniExpressExome-8v1-3_A DNA Analysis BeadChip was performed in the Life and Brain Center in Bonn, Germany. The DNA was processed according to the manufacturer’s instruction using an automatized, LIMS controlled workflow and the arrays were finally scanned using an Illumina iScan bead arrays reader. Genotype calling and quality control of the samples were carried out jointly in all 1349 participants using Illumina’s GenomeStudio v2011.1 software suite. For rare variants, zCall (threshold = 7) was applied [21]. Exclusions were due to low call rate, discordant sex information, related or duplicated individuals and divergent ancestry (deviating from CEU) [22,23,24], leaving a sample size of 1292 participants genotyped with the Illumina Infinium OmniExpressExome-8v1-3_A DNA Analysis BeadChip. Overall, we had data for *n* = 2269 samples available. Phasing and imputation were conducted using the Michigan Imputation Service [25]. The Haplotype Reference Consortium (release 1.1) served as reference panel [26]. Pre-phasing was applied using Eagle2 [27]. Imputation was carried out in four separate datasets (one for each chip or two for the HumanCoreExome-12v1-0_B chip) using minimac3 [25]. Imputed files were merged using bcftools [28], keeping the minimal R2 score. Afterwards, SNPs were filtered by R2 keeping those with values bigger than 0.6. Pre- and post-imputation tools (HRC-1000G-check-bim.v4.2.9, icv.1.0.5) for checking data quality were applied [29].

### 2.3. Determination of Desaturase Activities

Thirty milliliters of blood were obtained from EPIC-Potsdam participants during baseline examination. Plasma, serum, erythrocytes and buffy coat were stored at −80 °C. The erythrocyte membrane fatty acids were analyzed between February and June 2008. Thirty-two fatty acids were determined by gas chromatography and expressed as the percentage of total fatty acids present in the chromatogram [14]. Estimated D5D activity was determined as the ratio arachidonic acid (AA)/dihomo-γ-linolenic acid (DGLA) (20:4n-6/20:3n-6), and D6D activity as the ratio γ-linolenic acid (GLA)/linoleic acid (LA) (18:3n-6/18:2n-6), as done previously [1].

### 2.4. Statistical Analysis

We used the Statistical Analysis System (SAS) Enterprise Guide 7.1 with SAS version 9.4 (SAS Institute Inc., Cary, NC, USA) for data management and data preparation. For data filtering we used QCtool v1.4 and for GWAS we used SNPtest v2.5.2 [30]. We performed Mendelian Randomization analyses with R (version 3.5.2 (2018-12-20)) using the TwoSampleMR (v0.4.26) [31], the Mendelian Randomization (v0.4.1 and v0.4.2) [32], Radial MR (0.4) [33] and MVMR (0.2) [34] R packages. To visualize LD of the *FADS*-gene region and genetic instruments we utilized Plink v1.07, plink v1.90 [35] and Haploview v4.1 [36].

#### 2.4.1. Selecting Genetic Instruments in Genome-Wide Association Study

SNPs were filtered by SNP missing-rate (removed ≥ 0.05), minor allele frequency (MAF) (removed out of interval [0.05–0.5]) and Hardy–Weinberg equilibrium (removed -log10(*p*-value) ≥3). Genetic instruments were obtained from GWAS on natural log-transformed and standardized (mean = 0; SD = 1) estimated D5D and D6D using n~5,340,003 markers as exposures. We considered a *p*-value as genome-wide significant at *p* < 9.36 × 10^−9^ [0.05/5,340,003]. Suggestive significance threshold was defined as *p* < 1.00 × 10^−5^. We assumed an additive genetic model, adjusted for age at recruitment and sex. Variants were mapped to Ensembl annotation version 87 (GRCh37) [37] and we used the Ensembl Variant Effect Predictor for annotation [38].

#### 2.4.2. Mendelian Randomization

We conducted univariable two-sample MR studies using single desaturases as exposures on cardiometabolic outcomes. Effect estimates of the association between genetic instruments and desaturases were obtained from EPIC-Potsdam data and effect estimates of the SNP-T2DM and SNP-CAD associations were used from public summary GWAS data on T2DM [18] and CAD [19]. We selected gene-wide significant (*p* < [0.05/169]) instrumental variables and performed clumping according to LD. Therefore, SNPs within a window of 10,000 kb and being in LD as defined by *R*^2^ ≥ 0.3 were removed (Figure 2). The SNP with the lowest *p*-value was retained. Within the MR analysis we accounted for their correlation among each other estimated in 502 European samples from 1000 Genomes phase 3 [39]. We repeated the analysis by additionally including independent genome-wide significant hits (*p* < 9.36 × 10^−9^) that were not located within the *FADS* gene region (with a clumping-threshold of 0.001).

Data were harmonized for the direction of effects between exposure and outcome associations and palindromic SNPs were excluded. We used an inverse variance weighted (IVW) meta-analysis of SNP specific Wald ratios (SNP-outcome estimate divided by SNP-exposure estimate) using random effects, to obtain causal estimates for the desaturase activities on T2DM or CAD risk. MR-Egger method [40] was used in sensitivity analyses. Heterogeneity was assessed by the Cochran’s Q statistic and we performed RadialMR [33] to identify outliers with the largest contribution to the Q statistic. Outlying genetic instruments were removed and the data were re-analyzed. Radial MR analysis was conducted using second order weights and an α level of 0.05 or 0.1.

We applied a multivariable MR approach [15] combining both desaturases (see Figure 1b). First, we selected all gene-wide significant instruments for each estimated desaturase activity. This combined list was again pruned with *R*² < 0.3 leaving ten independent instruments. We obtained the SNP effects on the other desaturase activity and vice versa. Extraction of the outcome GWAS results and data harmonization was conducted as described for the univariable MR. We applied the IVW multivariable MR performing multivariable weighted linear regression with the intercept term set to zero and accounting for the correlation structure among the ten *FADS* variants. We additionally included genome-wide significant hits (*p* < 9.36 × 10^−9^) that were not located within the *FADS* gene region. As sensitivity analyses, we used multivariable MR-Egger for two-sample summary data [41]. We calculated F-statistics to evaluate the presence of weak instruments within the multivariable MR analysis and adjusted for those by minimizing the Q-statistic allowing for heterogeneity using “qhet_mvmr” function with 1000 bootstrap iterations from the MVMR package [34]. Phenotypic correlation was obtained from EPIC-Potsdam data. To account for horizontal pleiotropic effects by other pathways than considered in the multivariable approach, we used Causal Analysis Using Summary Effect estimates (CAUSE) [42].

#### 2.4.3. Investigation of LD within the FADS-Gene Cluster

To exclude possibly confounding by LD, we restricted the MR analysis for D6D on variants that are located in *FADS2* only and are not in LD (*R*² < 0.45) with SNPs located in *FADS1-FADS2* LD Block. The selection was based on a LD plot of all 143 D6D GWAS Hits available within the *FADS*-gene region in EPIC-Potsdam. For D5D, we included only genome-wide hits outside from *FADS* due to the strong LD of *FADS1* variants with variants in *FADS2*.

## 3. Results

### 3.1. Selection of Genetic Instruments

GWAS were conducted in EPIC-Potsdam with 1853 participants. Baseline characteristics are illustrated in Table 1.

Genetic variants within the *FADS* gene region were strongly associated with estimated D6D (Appendix A, Appendix A) and D5D activity (Appendix A, Appendix A). For D6D, we identified 143 FADS-gene-wide significant (*p* < [0.05/169]) variants of which after LD clumping seven independent variants (*R*² < 0.3) remained (Appendix A, Appendix A). All of them were available within DIAGRAM and one (rs174607) could not be found within CAD outcome data. However, this SNP (rs174607) was labeled as palindromic variant and was therefore excluded also from the diabetes analysis (Appendix A).

Regarding D5D, 160 SNPs showed *FADS*-gene-wide significance (*p* < [0.05/169]) resulting in 11 independent variants (*R*² < 0.3) (Appendix A, Appendix A). All of them were available within DIAGRAM and one (rs174607) was not available for CAD outcome data. Two variants (rs174607, rs174565) were excluded from the diabetes analysis because of being palindromic and are therefore not presented in Appendix A.

Furthermore, our GWAS identified two novel loci associated with D5D activity at genome-wide significance level: an intronic variant (rs2608073) at chromosome 3 within the *RP11-372E1.4* locus and an intronic variant (rs11644601) at the *RRN3*; *PDXDC1* locus at chromosome 16 (Appendix A and Appendix A).

### 3.2. Mendelian Randomization

#### 3.2.1. Univariable Mendelian Randomization

##### Causal Estimates for Desaturase Activities on Risk of Type 2 Diabetes

The IVW estimate indicated a positive total effect of D6D (odds ratio (OR) [95% confidence interval (CI)] = 1.08 [1.06–1.09]) on T2DM (Table 2, Appendix A). The results showed no indication of heterogeneity between the SNPs (Cochran’s Q value = 7.17, *p* = 0.21). The MR-Egger regression showed a positive intercept 0.012 (SE = 0.009) *p* = 0.185) and OR (1.04 [0.98–1.10]) (Appendix A), not supporting directional horizontal pleiotropy with regards to T2DM.

For D5D, we found a positive total effect (OR = 1.03 [1.01–1.04]) on T2DM (Table 2, Appendix A). There was evidence for heterogeneity among the SNPs (Cochran’s Q value = 28.25, *p* = 0.0004). The MR-Egger regression showed a positive intercept (0.011 (SE = 0.011), *p* = 0.306) and no association (OR = 1.00 [0.98–1.05]) (Appendix A), not indicating presence of directional horizontal pleiotropy. Based on the Radial MR, we identified two outlying SNPs (rs174602, rs508768) for the D5D—T2DM analysis (Appendix A). After outlier removal, inverse variance weighted and MR-Egger effect estimates were largely unchanged (Appendix A; Appendix A); however, heterogeneity was not present anymore (Cochran’s Q value = 6.55, *p* = 0.36).

##### Causal Estimates for Desaturase Activities on Risk of Coronary Artery Disease

We found evidence for a causal total effect of D6D (OR = 1.06 [1.02–1.11]) on the risk of CAD (Table 2, Appendix A) with no indication of heterogeneity. The MR-Egger estimates were lower and not significant anymore compared to the IVW estimates (Appendix A). For D5D, we found a positive total effect (OR = 1.03 [1.01–1.05]) on the risk of CAD (Table 2, Appendix A) with no indication of heterogeneity. However, MR-Egger estimates were lower than the ones from the IVW method and not significant. Furthermore, the MR-Egger test indicated directional horizontal pleiotropy for D5D and CAD, pointing to a biased IVW estimate (Appendix A). When we excluded one possibly outlying SNP (rs61897792) the indication for directional horizontal pleiotropic effects for D5D and CAD disappeared (Appendix A; Appendix A).

##### Sensitivity Analyses

In sensitivity analyses, we additionally included genome-wide significant instruments (*p* < 9.36 × 10^−9^) for D5D. For both endpoints, there was evidence for heterogeneity between the SNPs (T2DM: Cochran’s Q value = 31.15, *p* = 0.0006; CAD: Cochran’s Q value = 20.46, *p* = 0.03). The indication for directional horizontal pleiotropy with regards to CAD remained (Appendix A). The observed positive associations for T2DM (OR = 1.04 [1.02–1.06]) and CAD (OR = 1.03 [1.01–1.06]) did not change (Table 2, Appendix A). Exclusion of the previous two outlying SNPs (rs174602, rs508768) influenced the IVW effect estimate for T2DM only marginally (Appendix A; Appendix A). Exclusion of one possibly outlying SNP (rs61897792) did not change the IVW effect estimates for CAD, but weakened the indication for directional horizontal pleiotropic effects with regards to D5D and CAD (Appendix A; Appendix A). The CAUSE method suggested indication for causal association for both desaturases with T2DM and CAD (Appendix A).

#### 3.2.2. Multivariable Mendelian Randomization

##### Estimates for Causal Direct Effects of Desaturase Activities on Risk of Type 2 Diabetes and Coronary Artery Disease

All independent *FADS*-gene-wide significant variants (*p* < [0.05/169]) from both desaturases were combined leading to ten variants of which all were available within the outcome GWAS (Appendix A). In the multivariable MR approach, we could not precisely allocate the causal effect on T2DM to one or the other desaturase. While for D6D, we still found a small not significant direct effect on the risk of T2DM (OR = 1.03 [0.94–1.12]); for D5D, there was no direct causal effect (OR = 1.00 [0.96–1.04]). For CAD, the total causal effect of D6D was fully attenuated in multivariable MR (OR = 1.00 [0.93–1.07]) (Table 2); however, a significant direct causal effect of D5D on the risk of CAD could be observed (OR = 1.04 [1.01–1.08]) (Table 2). Though, we had indication for weak instruments with F-statistic < 10 (D5D = 9.91, D6D = 2.23). We repeated the multivariable MR by accounting for weak instruments and received comparable direct effect estimate for D5D (OR = 1.04 [1.01–1.15]) (Table 2).

##### Sensitivity Analyses

Additional inclusion of genome-wide significant instruments did not change the observation of no significant direct causal effect of D6D on the investigated outcomes while the direct effect of D5D on CAD was slightly attenuated (OR = 1.03 [0.99–1.06]). However, the F-statistics were higher (D5D = 14.41, D6D = 3.06). When we accounted for weak instruments, the estimates did not largely change (Table 2).

The multivariable MR-Egger regression showed intercepts that were close to zero, not indicating presence of directional horizontal pleiotropy (Appendix A). Exclusion of outlying instruments (rs174602, rs508768) did not markedly influence estimates, however, they lost precision (e.g., direct effect of D5D on CAD using only *FADS*-instruments: 1.03 [0.97–1.09], *p*-value = 0.336) (Appendix A). Exclusion of an additional outlying instrument (rs61897792) did not further change effect estimates (Appendix A).

#### 3.2.3. Investigation of Confounding by Linkage Disequilibrium

Finally, when we excluded instruments from *FADS2* that are in LD with *FADS1*-SNPs to rule out confounding by LD (Appendix A), we observed considerably stronger positive total effect of D6D on T2DM (1.12 [1.06–1.18]) and CAD (1.12 [1.04–1.21]) (Table 3). While the effect estimate was comparable in multivariable MR for CAD, these analyses lost precision and no significant direct effects were observed. When we excluded one SNP (rs174602) from the *FADS2* instruments that was reported to be associated with *FADS1* expression [43], the results did not change (Appendix A) When we included only genome-wide significant hits outside from *FADS* for D5D, we still observed positive total effect on T2DM (1.04 [0.99–1.08]) and CAD (1.04 [0.98–1.11]), although not significant. Multivariable MR did not indicate direct effects of D5D on risk of T2DM or CAD (Table 3). 

## 4. Discussion

Within this study, we applied a two-sample MR approach to investigate causal effects of estimated desaturase activities on the risk of T2DM and CAD. Our univariable MR indicated that PUFA-generating desaturases are causally involved in the development of both, T2DM and CAD. For D6D these effects became more prominent if we accounted for confounding by LD in our genetic instruments. However, due to low precision, attribution of the direct causal effect on T2DM to D6D activity remains suggestive. For CAD, results from the multivariable MR suggested a direct effect only for D5D activity, but we were not able to exclude confounding by LD. It was therefore not possible to disentangle the causal impact of the single desaturases on CAD risk.

### 4.1. D6D and Risk of Type 2 Diabetes and Coronary Artery Disease

The D6D catalyzes the desaturation of linoleic acid (LA) that shows inverse association with cardiovascular disease mortality [44] and cardiovascular events [45]. The product of the D6D, γ-linolenic acid (GLA) and higher D6D activity itself are positively associated with T2DM [2,14]. Our results from univariable MR are in line with a previous MR study reporting that a genetically determined high D6D activity [14] predicted higher T2DM risk. Also, low levels of LA, representing a D6D substrate, were associated with increased risk of T2DM in recent MR analysis [12]. However, selecting instruments for D6D is affected by the strong LD in the *FADS* gene cluster, where one LD block spans *FADS1* and parts of the *FADS2* gene [1]. Therefore, genetic instruments for D6D (from *FADS2*) cannot be easily separated from those for D5D (*FADS1*).

We tried to overcome this problem of confounding by LD for D6D by restricting the selected instruments to those from the *FADS2* gene that are not in LD with variants from *FADS1*. This sensitivity analysis revealed stronger total effects of D6D on both T2DM and CAD, which further supports a causal role of D6D. Still, due to the considerable loss of precision in the multivariable MR the allocation of the T2DM-risk augmenting effect of D6D activity exclusively to those PUFAs being substrate or product of D6D activity (LA and GLA) remains relatively uncertain. More likely, downstream formation of PUFAs from GLA mediate an important part of the total effect of D6D.

### 4.2. D5D and Risk of Type 2 Diabetes and Coronary Artery Disease

Regarding D5D, alignment of our results from univariable MR with non-genetic observational studies is more complex. D5D, catalyzes the conversion of DGLA to arachidonic acid (AA). Our univariable MR suggested links of higher D5D with higher risk of T2DM. This is in line with a recent MR study showing higher T2DM risk with higher levels of AA [12]. In contrast to that, observational studies relate higher estimated D5D activity and higher AA levels to lower T2DM risk [2,14]. A direct link of D5D with T2DM was, however, not observable in multivariable MR. Also, instruments for D5D (*FADS1*) are largely confounded by LD with *FADS2* variants and we were not able to overcome this LD problem by using instruments outside the *FADS* region. Therefore, our finding of a higher T2DM risk among participants with genetically high D5D activity in univariable MR reflects most likely the genetic linkage with D6D rather than an independent effect.

In contrast to T2DM, the observational evidence that links higher estimated D5D activity to lower cardiovascular mortality and CHD risk is rather limited [4,5], where an inverse association of high D5D activity was limited to participants who were homozygous for the major allele (AA genotype; rs174547) [5]. A previous MR on fatty acids showed a positive effect of higher AA levels on CVD risk [13], although this study did not evaluate fatty acid ratios as estimate of D5D activity. While our univariable MR analysis supports a total causal effect of D5D activity on CAD, interpreting these findings as reliable evidence for causal effects is limited. We had significant MR-Eggers test, indicating that directional (unbalanced) horizontal pleiotropy was present or that the InSIDE assumption was violated or both [46]. We did not identify outliers, that could explain this finding. Furthermore, although we identified a significant direct effect of D5D on CAD when using only *FADS*-instruments, this result was sensitive to inclusion of genome-wide instruments and outlier exclusions. As pointed out above, it is virtually impossible to account for confounding by LD. Restricting our analysis to variants outside of the *FADS*-gene region hampered precision of the D5D estimates to an extent that made them uninformative. Hence, we do not consider effect of D5D on CAD risk to be robust and will therefore restrain from causal interpretation.

### 4.3. Biological Mechanisms

PUFAs play a crucial role in cell membrane fluidity and thereby influence insulin receptor binding affinity and endothelial function [47] which might represent one mechanism how altered desaturation of fatty acids influences the risks of T2DM [48] and CAD [49]. Additionally, PUFAs affect transcription factors such as sterol regulatory element binding protein 1 and peroxisome proliferators activating receptors regulating genes involved in control of lipid flux into and out of the liver, which is important in terms of reducing hepatic lipid accumulation and hence hepatic insulin resistance [50]. Furthermore, PUFAs are substrate for the formation of various lipid-related metabolites, e.g., eicosanoids, leukotriens, prostaglandins, thromboxanes, lipoxins, endo-cannabinoids, or resolvins, which themselves are highly bioactive [47]. AA-related eicosanoids may increase vasoconstriction and platelet activation and aggregation, promoting atherosclerotic plaque formation as pathological processes in the development of CAD [51].

### 4.4. Strengths and Limitations

By using instruments from the *FADS* region, we used well established genetic candidates from the genic region of the investigated exposures that explain up to ~30 % (rs174555 and D5D) of the variance in the traits (R² ranging from 5 to 10% for D6D and 1 to 30% for D5D). Although we had indication for weak instruments within the multivariable MR, we retrieved comparable estimates when we accounted for that. Besides genetic candidates, we also used genome-wide screen and identified two novel hits for D5D that have not been reported by previous GWAS on estimated desaturase activities [8,9]. Although our genome-wide significance cut-off might be too strict, we would not have found additional hits using the common threshold of *p* < 5 × 10^−8^. Furthermore, we wanted to be more conservative, as we did not replicate those findings in an independent population. Nevertheless, our MR results incorporating those genome-wide hits were comparable to the main analysis when using only *FADS* variants.

We minimized potential incorrect causal inference due to horizontal pleiotropy by restricting genetic instruments on those that plausibly act directly on the traits as they were located in the coding genes [52]. Furthermore, we corrected for correlated and uncorrelated horizontal pleiotropy using CAUSE method and still retained indication for causal effect of both desaturases and cardiometabolic outcomes. Additionally, there must be no LD with other variants that might influence the expression or activity of a different protein as this can reintroduce confounding by LD [52]. Similar to the pleiotropy situation, this would violate key assumptions of MR [50]. We addressed confounding by LD by accounting for the LD structure in the main analysis and by restricting genetic instruments to those that are not in LD with instruments for the respective other desaturase. Still, there might be criticism as the two instruments (rs498793, rs7118175) for D6D used in sensitivity analyses showed significant associations with D5D levels and might therefore be considered as not independent from *FADS1*; however, we did not identify genome-wide significant instruments for D6D outside of *FADS* which could have been used instead.

One limitation of our analysis lies in the exclusion of prevalent cases in the EPIC-Potsdam sample which was used for instrument selection and might have weakened the exposure betas. However, compared to the overall study sample this number was small and we aimed to exclude bias from reverse confounding due to the fact that prevalent cases might have different desaturase activity [52]. Also, our multivariable MR might be underpowered; however, we did no power calculation, as this is—to our knowledge—only developed for continuous outcomes [53]. Furthermore, we did not directly measure desaturase activities. In humans, D6D and D5D are mainly expressed in the liver [54,55]. However, direct measurement of liver desaturase activity would require liver biopsies which is not possible in population-based settings. Therefore, product-to-precursor ratios of fatty acids measured in blood fractions represent well-established surrogate markers of estimated liver desaturase activity in epidemiological research [1,56]. Another limitation refers to the selection of the outcome GWAS for CAD. Ideally, the exposure and outcome data should arise from the same population in terms of ancestry. For T2DM, this was fulfilled; however, for CAD, also Asian individuals contributed to the meta-analysis [19]. Nevertheless, this should not increase the likelihood of finding an association when there is none [57], hence our results on D6D and CAD might still be informative in terms of effect directions of causality [31].

## 5. Conclusions

In conclusion, our results suggest that D6D is causally linked to cardiometabolic risk. However, the effect is likely not directly due to its substrate or product fatty acids, but rather due to the downstream production of fatty acids (DGLA and AA) and their products resulting from high D6D activity. For D5D, our MR approach suggests a causal risk-increasing effect for T2DM and CAD risk; however, we were not able to fully rule out confounding by LD. Interventions that affect desaturase activities in PUFA metabolism may have an impact on the pathogenesis of cardiometabolic diseases.

## Figures and Tables

**Figure 1 nutrients-12-02261-f001:**
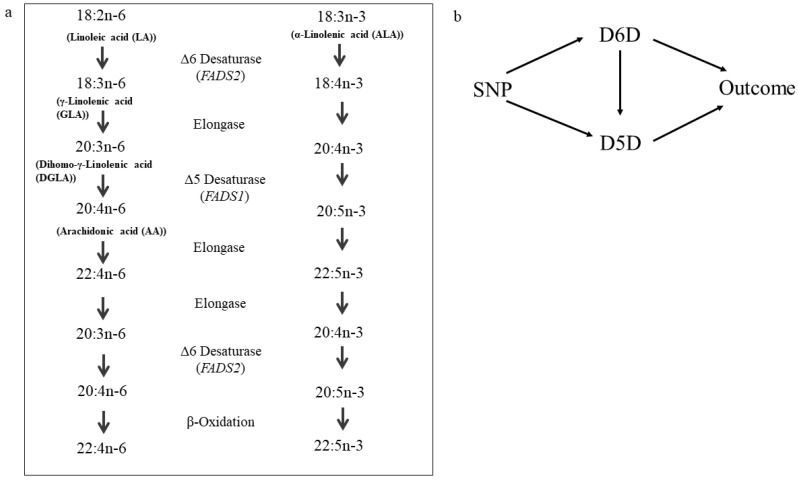
(**a**) Conversion of linoleic acid and α-linolenic acid to longer-chain n-6- and n-3 polyunsaturated fatty acids by the action of Δ6-desaturase, Δ5-desaturase and elongases. Adapted from Kröger et al. [1]. (**b**) Hypothesized relationship between genetic variants, Δ6-desaturase (D6D), Δ5-Desaturase (D5D) and the investigated outcomes type 2 diabetes and coronary artery disease. The arrow from D6D to D5D relies on the biologically underlying mechanisms as depicted in panel a; however, methodologically in the multivariable MR the arrow could be in any direction. Confounders are omitted from this figure for clarity. Pleiotropic effects by pathways other than D6D or D5D are not considered in this multivariable MR.

**Figure 2 nutrients-12-02261-f002:**
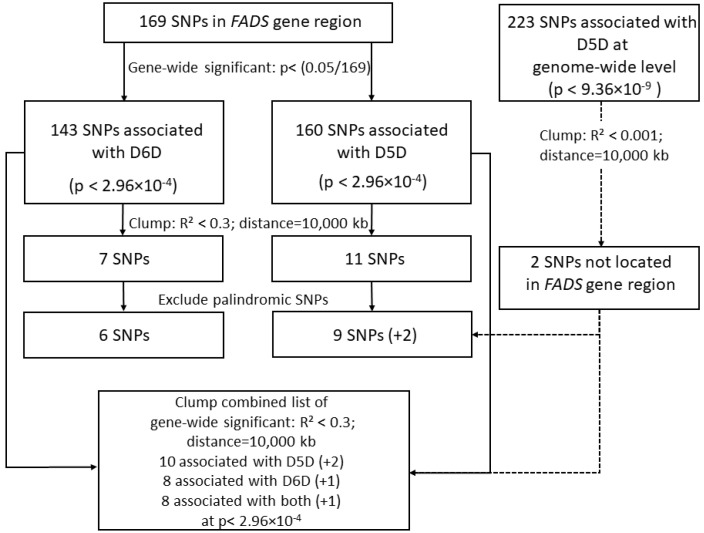
Flow chart of inclusion and exclusion of SNPs into the analysis. First, we restricted to the set of SNPs that were located within the *FADS* gene region (Chr 11: 61560452–61659523). Next, we clumped the GWAS results separately for D6D and D5D to identify independent SNPs for both traits for univariable MR approach. Finally, we clumped a combined list of SNPs for both traits for multivariable MR. This resulted in ten SNPs for D5D and eight SNPs for D6D at gene-wide significance level *p* < 2.96 × 10^−4^. There were eight SNPs that associated with both desaturases. All SNPs were included in the analysis. Additionally, two SNPs at genome-wide significance level for D5D were included.

**Table 1 nutrients-12-02261-t001:** Baseline characteristics, EPIC-Potsdam random sample.

	EPIC-Potsdam
N	1853
Sex (% men)	37.1
Age in years; median (interquartile range)	49.0 (15.5)
Waist circumference in cm; mean (SD)	85.3 (12.6)
Δ6-desaturase activity (18:3n-6/18:2n-6); median (interquartile range)	0.005 (0.003)
Δ5-desaturase activity (20:4n-6/20:3n-6); mean (SD)	8.80 (1.91)
Lipid medication (%)	3.72

SD, standard deviation.

**Table 2 nutrients-12-02261-t002:** Total and direct effects of estimated desaturase activities and risk of type 2 diabetes and coronary artery disease.

			T2DM			CAD		
		Method	N (SNPs) *	OR (95% CI)	*p*-Value	N (SNPs) †	OR (95% CI)	*p*-Value
**D6D**	instruments from *FADS*	IVW	6	1.08 (1.06–1.09)	<0.001	6	1.06 (1.02–1.11)	0.008
		MVIVW	10	1.03 (0.94–1.12)	0.528	10	1.00 (0.93–1.07)	0.971
		MVIVW ‡	10	1.03 (0.99–1.16)		10	1.01 (0.91–1.12)	
	instruments from *FADS* and genome-wide hits	MVIVW	12	1.03 (0.95–1.10)	0.514	12	1.00 (0.95–1.06)	0.907
		MVIVW ‡	12	1.01 (0.94–1.10)		12	1.00 (0.96–1.15)	
**D5D**	instruments from *FADS*	IVW	9	1.03 (1.01–1.04)	<0.001	9	1.03 (1.01–1.05)	0.017
		MVIVW	10	1.00 (0.96–1.04)	0.824	10	1.04 (1.01–1.08)	0.021
		MVIVW ‡	10	1.02 (0.98–1.04)		10	1.04 (1.01–1.15)	
	instruments from *FADS* and genome-wide hits	IVW	11	1.04 (1.02–1.06)	<0.001	11	1.03 (1.01–1.06)	0.017
		MVIVW	12	1.00 (0.96–1.03)	0.845	12	1.03 (0.99–1.06)	0.108
		MVIVW ‡	12	1.02 (0.99–1.05)		12	1.04 (0.99–1.07)	

CAD, coronary artery disease; CI, confidence interval; D5D, delta-5-desaturase; D6D, delta-6-desaturase; IVW, inverse variance weighted method; MVIVW, multivariable inverse variance weighted method; OR, Odds ratio; T2DM, type 2 diabetes. * Of the 11/7 SNPs associated with D5D/D6D, 11/7 were available in the GWAS of T2DM [18]. After harmonization and removal of palindromic SNPs with intermediate allele frequencies, 9/6 SNPs were included in the MR analysis on T2DM. For MVMR a combined set of 10 SNPs was used. † Of the 11/7 SNPs associated with D5D/D6D, 10/6 were available in the GWAS of CAD [19]. After harmonization and removal of palindromic SNPs with intermediate allele frequencies, 9/6 SNPs were included in the MR analysis on CAD. For MVMR a combined set of 10 SNPs was used. ‡ adjusting for weak instruments in MVMR, but not for correlation structure between instruments.

**Table 3 nutrients-12-02261-t003:** Total and direct effects of estimated desaturase activities and risk of type 2 diabetes and coronary artery disease accounting for confounding by LD.

		T2DM				CAD			
	Method	N (SNPs)	OR (95% CI)	*p*-Value	Intercept (SE), *p*-Value	N (SNPs)	OR (95% CI)	*p*-Value	Intercept (SE), *p*-Value
**D6D**	MR-Egger	3 †	0.87 (0.55–1.37)	0.538	0.050 (0.045), 0.267	3 †	1.26 (0.21–7.70)	0.804	−0.022 (0.176), 0.902
	IVW *	3 †	1.12 (1.06–1.18)	<0.001		3 †	1.12 (1.04–1.21)	0.002	
	MVIVW *	3 †	0.74 (0.34–1.62)	0.453		3 †	1.12 (0.34–3.69)	0.850	
**D5D**	IVW *	2 ‡	1.04 (0.99–1.08)	0.087		2 ‡	1.04 (0.98–1.11)	0.236	
	MVIVW *	2 ‡	1.01 (0.95–1.07)	0.790		2 ‡	1.00 (0.93–1.08)	0.996	
	MVIVW *	3 †	1.29 (0.75–2.21)	0.362		3 †	1.00 (0.48–2.09)	1.000	

* Fixed effect model; † including only *FADS1*-independent (R² < 0.45) *FADS2* variants (rs174602, rs498793, rs7118175); ‡ including only genome-wide hits for D5D (rs2608073, rs11644601).

## Data Availability

The EPIC-Potsdam datasets analyzed during the current study are not publicly available due to data protection regulations. In accordance with German Federal and State data protection regulations, epidemiological data analyses of EPIC-Potsdam may be initiated upon an informal inquiry addressed to the secretariate of the Human Study Center (Office.HSZ@dife.de). Each request will then have to pass a formal process of application and review by the respective PI and a scientific board. Summary-level data for genetic associations with the cardiovascular disease and type 2 diabetes outcomes have been contributed by the CARDIoGRAMplusC4D (http://www.cardiogramplusc4d.org/data-downloads/; accessed on 12 August 2019) and DIAGRAM consortia (http://diagram-consortium.org/downloads.html; accessed on 8 August 2019). The authors thank all investigators for sharing these data.

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
