# Peer review of "Desaturase Activity and the Risk of Type 2 Diabetes and Coronary Artery Disease: A Mendelian Randomization Study"

_nutrients, 2020, doi:10.3390/nu12082261_

Round 1
Reviewer 1 Report
This manuscript examines the causal effect of delta 5-desaturase (D5D) and delta 6-desaturase (D6D), which are enzymes involved in the metabolism of polyunsaturated fatty acids, on the risk of type 2 diabetes and coronary artery disease, using a Mendelian randomization approach.
The authors have performed a thorough analysis and conclude that D6D is likely causally associated with cardiometabolic risk but that there is not enough evidence to say the same for D5D. Overall, this is a comprehensive piece of work and my suggestions for improvement are given below.
- The IV-exposure association was obtained from a GWAS carried out in a relatively small sample, where SNPs in the FADS region were replicated and a couple of new signals were uncovered. The latter, however, have not been replicated in other samples, so their relevance as instruments may be in question.
- It would be good to have an estimate of instrument strength as well as of statistical power for the MR analyses.
- The authors could also run two-sample MR using data from existing GWAS on D5D and D6D, which seem to have identified associated SNPs outside of the FADS region, and see whether they obtain similar results.
- MR-Egger can also be considered a sensitivity analysis.
- An additional MR analysis including suggestive GWAS associations could have been run as a sensitivity analysis as well.
- I find the LD R2 cut-off for selecting SNPs 0.3 somewhat high. This still represents quite a strong correlation. Why was this threshold picked?
- Similarly, LD R2 cut-off of 0.45 for the `Investigation of LD within the FADS-gene cluster’ also seems high to be able to exclude the influence of FADS1.
- I am not convinced that the main conclusion of this work is that D6D, but not D5D, is causally linked to cardiometabolic traits. Based on the univariable MR analysis we could say that about both desaturases, although for D5D and CAD there was some evidence of directional pleiotropy. The multivariable MR only showed evidence of a causal effect of D5D on CAD, which was weakened by outlier exclusions, yet it remained positive. I would not put a lot of emphasis on the contribution of the newly discovered genome-wide hits. While it is true that there is high LD in the FADS region that makes it difficult to isolate the different signals, the authors also reported that the multivariable MR accounted for the correlation structure of the ten FADS variants. So it seems to me that the most interesting result is that of D5D and CAD.
Minor comments
- page 2, line 52: define CHD.
- page 2, line 57: FADS1 and FADS2 should be in italics throughout the manuscript.
- page 3, line 98: explain SOFT CAD.
- page 3, line 114: explain “divergent ancestry” (presumably non-European).
- page 4, lines 124 and 126: start sentences spelling out the numbers.
- page 4, lines 128-129: define AA, GLA and LA.
- page 4, lines 134-136: sentence is confusing, please rephrase.
- page 4, line 139: “minor” allele frequency.
- page 5, line 165: 2.96x10-4.
- page 5, line 186: write LD R2 consistently (either always with R or with r).
- page 6, lines 202-203: should be Supplementary Table 1.
- page 8, lines 267-268: for D6D report OR for FADS instruments only, same as is being done for D5D.
- page 8, line 275: “lost significance” does not sound right. In general, I would not focus too much on significance.
- page 11, line 357: eicosanoids.
- Table 3: please show the MR-Egger intercept estimates and p-values as well.
- Supplementary Tables 1 & 2: clarify that EAF is %.
- Supplementary Table 2: rs61897792 is not genome-wide significant according to the cut-off reported in Methods.
- Supplementary Figure 2: it should say the horizontal red line and the horizontal blue line and not the vertical.
- Supplementary Figure 3: Please include in the graphs all the FADS SNPs used, show the extension of FADS1 and FADS2 and report the method used for block definition.
Reviewer 2 Report
The authors have conducted a well-designed and thorough two-sample Mendelian randomization (MR) analysis to assess the causality of desaturase activity on type 2 diabetes and coronary artery disease. They found that both D6D and D5D were positively associated with T2DM risk, and D6D with CAD risk, by univariable MRs. Their multivariable MRs suggested that high D6D cause cardiometabolic conditions and D5D-T2DM correlation come from LD with D5D. The writing is excellent, and the statistical analysis is overall rational. However, I have several comments/suggestions as below:
- As mentioned in Figure 1, current multivariable MR did not consider pleiotropic effects by other pathways. The authors may acknowledge this limitation and consider solution(s) such as using the Causal Analysis Using Summary Effect estimates (CAUSE) approach (PMID: 32451458).
- Your genome-wide significance level may be too conserved. I would suggest you use P<5e-8 instead to identify D6D and D5D loci in your individual data.
- In Figure 2 of Page 5, two approaches were used to select instrumental SNPs. One is from those gene-wide significant SNPs in the FADS gene cluster region, select independent SNPs by LD clumping. Why you use 10 Mb for clumping? Is it reasonable to select independent SNPs by association results? For example, using stepwise linear regression models to choose independent SNPs from 169 candidates, and further filter the selected SNPs by removing those genetically correlated ones with LD r2>0.1.
- In Supplementary Tables 1 and 2, it would be better to provide harmonized summary data (i.e., the beta values for D6D or D5D in the same direction, all >0).
- In the text of Page 5, line 168, palindromic SNPs were excluded. What are their MAF? If MAF<0.45 or less, such SNPs may be able to be harmonized manually.
- On Page 6, lines 209-211, the two SNPs identified through GWAS are imputed? If so, what are the imputation quality scores?
- On Page 11, lines 363, 364 the author mentioned ~30% variance explained by the instrumental SNPs, what is the power for univariable MR?
Minor. In Figure 2, “-4” of “2.92x10-4” should be in superscript.
